

# Soil respiration and its $Q_{10}$ response to various grazing systems of a typical steppe in Inner Mongolia, China

Cheng Nie[1], Yue Li[2], Lei Niu[1], Yinghui Liu[1], Rui Shao[3], Xia Xu[1] and Yuqiang Tian[1]

[1] State Key Laboratory of Earth Surface Processes and Resource Ecology, Faculty of Geographical Science, Beijing Normal University, Beijing, China
[2] South China Botanical Garden, Chinese Academy of Sciences, Guangzhou, China
[3] Department of Geography, Ghent University, Ghent, Belgium

Corresponding author
Yinghui Liu, lyh@bnu.edu.cn

## ABSTRACT

**Background:** As one of the important management practices of grassland ecosystems, grazing has fundamental effects on soil properties, vegetation, and soil microbes. Grazing can thus alter soil respiration (Rs) and the soil carbon cycle, yet its impacts and mechanisms remain unclear.

**Methods:** To explore the response of soil carbon flux and temperature sensitivity to different grazing systems, Rs, soil temperature (ST), and soil moisture (SM) were observed from December 2014 to September 2015 in a typical steppe of Inner Mongolia under three grazing systems: year-long grazing, rest-rotation grazing, and grazing exclusion. In addition, plant aboveground and root biomass, soil microbial biomass and community composition, and soil nutrients were measured during the pilot period.

**Results:** Soil respiration was significantly different among the three grazing systems. The average Rs was highest under rest-rotation grazing (1.26 $\mu mol \cdot m^{-2} \cdot s^{-1}$), followed by grazing exclusion (0.98 $\mu mol \cdot m^{-2} \cdot s^{-1}$) and year-long grazing (0.94 $\mu mol \cdot m^{-2} \cdot s^{-1}$). Rs was closely associated with ST, SM, potential substrate and root, and soil microbe activity. The effects of grazing among two grazing systems had generality, but were different due to grazing intensity. The root biomass was stimulated by grazing, and the rest-rotation grazing system resulted in the highest Rs. Grazing led to decreases in aboveground and microbial biomass as well as the loss of soil total nitrogen and total phosphorus from the steppe ecosystem, which explained the negative effect of grazing on Rs in the year-long grazing system compared to the grazing exclusion system. The temperature sensitivity of Rs ($Q_{10}$) was higher in the rest-rotation and year-long grazing systems, likely due to the higher temperature sensitivity of rhizosphere respiration and higher "rhizosphere priming effect" in the promoted root biomass. The structural equation model analysis showed that while grazing inhibited Rs by reducing soil aeration porosity, ground biomass and SM, it increased $Q_{10}$ but had a lower effect than other factors. A better understanding of the effects of grazing on soil respiration has important practical implications.

## INTRODUCTION

The carbon stored in soil and vegetation plays an important role in the carbon pool of a terrestrial ecosystem. The terrestrial ecosystem carbon cycle includes carbon sources and sinks in the air, plants, and soil communities (*Piao et al., 2009*). Carbon is accumulated by plant photosynthesis and released by soil respiration (Rs). Rs is fundamental to the global carbon cycle because even small variations in Rs can influence the atmospheric $CO_2$ concentration and soil organic carbon (SOC) storage (*Zhang et al., 2015b*). Studying Rs and its temperature sensitivity can contribute to the understanding of how the terrestrial C cycle will respond to rising temperatures under global climate change.

Temperate grassland ecosystems make up 32% of the earth's terrestrial ecosystems (*Adams et al., 1990*). Grassland soil $CO_2$ emissions into the atmosphere contribute substantially to the global carbon cycle (*Craine & Wedin, 2002*). Rs reflects the metabolic activities of roots and soil microbes. It has been acknowledged that Rs can be affected by soil temperature (ST), soil moisture (SM) (*Raich & Schlesinger, 1992*), and soil element content (*Shi, Gao & Jin, 2015*). As one of the important management practices in grassland ecosystems, grazing is suggested to affect the quantity of carbon released from the soil (*Kioko, Kiringe & Seno, 2012*; *Jiao et al., 2012*), despite current evidence suggesting no clear relationship between grazing and the carbon budget in grassland ecosystems (*Liu, Zang & Chen, 2016*; *Wang et al., 2017*). It is suggested that grazing may decrease soil $CO_2$ flux during the growing season by reducing canopy photosynthesis and carbon translocation to the rhizosphere (*Bremer et al., 1998*). In addition, grazing can also decrease Rs by destroying the soil microenvironment and inhibiting microbial growth (*Zhao et al., 2017*). In addition, grazing may promote nutrient cycling and thus provide more respiratory substrates for microbes (*Stark, Strommer & Tuomi, 2002*; *Allard et al., 2007*). Grazing exclusion, one of the most common grazing systems for grassland restoration in Inner Mongolia that is usually achieved by fencing, has been found to promote Rs due to its positive effect of increasing root biomass and SOC (*He et al., 2008*). However, *Chen et al. (2016)* reported that Rs was decreased under grazing exclusion due to reduced ST and increased SM. Rest-rotation grazing is indicated to be a good system for balance between economic development through animal husbandry and the protection of grassland ecosystems. This system involves a period of rest, or short-term grazing exclusion, during the growing season. Previous studies have shown that rest-rotation grazing could increase SOC, belowground biomass and soil water storage (*Zhang et al., 2015a*). A meta-analysis of the responses of the soil microbial community size and Rs to grazing showed that only heavy grazing intensity significantly reduced Rs by inhibiting soil microbes (*Zhao et al., 2017*). Therefore, the impact of rest-rotation grazing on Rs may be different from grazing exclusion or year-long grazing.

In East Asia, most grassland ecosystems are subjected to varying degrees of grazing pressure and climate warming. The temperature sensitivity of Rs ($Q_{10}$, the increase in the $CO_2$ release for every 10 °C increase in temperature) could predict the response of Rs to climate change. $Q_{10}$ can be affected by a variety of factors, including ST and moisture, root activity, quality, and quantity of substrate, and microbial community size

(*Curiel Yuste et al., 2004*; *Zhang et al., 2015b*). The influence of grazing on $Q_{10}$ is still controversial (*Paz-Ferreiro et al., 2012*; *Chen et al., 2016*). Grazing can debase the $Q_{10}$ of autotrophic respiration by reducing aboveground and root biomass of vegetation (*Xu & Qi, 2001*). This decrease in aboveground biomass and litter input to soil will then reduce respiratory substrates and consequently decrease the $Q_{10}$ of heterotrophic respiration under grazing systems (*Cao et al., 2004*). However, previous studies also suggest that livestock may stimulate root exudation and litter decomposition, which can increase the quality of soil organic matter (SOM) and thus promote the $Q_{10}$ of heterotrophic respiration (*Zhao et al., 2016*). Furthermore, the $Q_{10}$ of fine root respiration is higher than that of microbes (*Boone et al., 1998*). Therefore, grazing may alter the $Q_{10}$ of Rs by changing the composition of Rs.

Grazing and enclosure are the representative land management modes of the typical steppe, but the effects of different management methods on the carbon cycle in grassland ecosystems have not been clear. We designed and implemented an experimental study to improve the understanding of the impact of different grazing systems on carbon cycling and the soil carbon budget of grasslands in the Inner Mongolian plateau. This study focuses on the effects of grazing on typical steppe, which may vary. Combined with soil, vegetation, or microbial indicators, the purpose of this study is to further explore how grazing affects Rs and its $Q_{10}$. The results will help to assess the ecological effects and climate responses of typical steppe under different grazing systems. The objectives of the study were to (1) reveal the annual dynamics of Rs and its relationship to environmental variables (i.e., ST and SM) and (2) explain the effects of different grazing systems on Rs and determine the $Q_{10}$ values of Rs in different grazing systems. Our hypotheses were as follows: (1) Grazing reduces Rs by inhibiting the growth and activity of the plant and soil microbial community; (2) The effect of grazing on Rs may be different between rest-rotation grazing and year-long grazing systems; and (3) Grazing alters $Q_{10}$ by changing the proportion of root respiration in Rs.

## MATERIALS AND METHODS

### Study area

The study area is located in Duolun County (41°46′–42°39′N, 115°50′–116°55′E), Inner Mongolia Autonomous Region, China. The annual mean air temperature is 3.3 °C, with a maximum mean temperature of 19.9 °C in July and a minimum mean temperature of −15.9 °C in January. The mean annual precipitation is 399 mm, and 88% of the precipitation occurs during the growing season extending for approximately 150 days from May to September (*Miao et al., 2009*). The vegetation in the study area is classified as typical steppe and dominated by *Stipa krylovii* Roshev., *Leymus chinensis* (Trin.) Tzvelev., *Artemisia frigida* Willd., *Agropyron cristatum* (L.) Gaertn., *Allium bidentatum* Fisch., and *Cleistogenes squarrosa* (L.) Keng. (*Li et al., 2013*). SOC ranged from 1% to 3%, and soil total nitrogen (TN) ranged from 0.1% to 0.3% in this study.

We selected three established grazing systems at the Duolun Restoration Ecology Research Station (part of the Institute of Botany, Chinese Academy of Sciences, 42°02′N, 116°17′E), which is managed by the Institute of Botany, Chinese Academy of Sciences. The "year-long grazing system" had been grazed by 4.75 sheep ha$^{-1}$ year$^{-1}$ since 2002.

The "rest-rotation grazing system" had been grazed by 2–4.5 sheep $ha^{-1}$ $year^{-1}$ since 2004, with an annual rest period from 10 March to 10 June. In the "grazing exclusion system," livestock had been completely excluded since 2002. We randomly located three experimental plots, each $10 \times 5$ m, within the area of each grazing system.

Field experiments were approved by the Duolun Restoration Ecology Research Station (part of the Institute of Botany, Chinese Academy of Sciences).

### Soil respiration, temperature, and moisture

Soil respiration was measured on each plot in late December 2014 and during the 2015 growing season in late March, May, July, August, and September. To measure Rs, one PVC collar (20 cm in diameter and seven cm in height) was set up at a random location within each plot. The collars were inserted into the soil where aboveground biomass had been removed at least 24 h before the Rs measurement was to be taken in order to exclude $CO_2$ released by aboveground vegetation. The collars were maintained in the plots for the six measurements to minimize the effect of soil heterogeneity. The daily dynamics of Rs were captured by hourly measurement for 24 h. Each measurement lasted 3 min, including 120 s of observation in which the first 30 s was excluded due to the change in air pressure, and 30 s of pre- and post-purge time. Respiration rates were measured using a LI-COR 8150 automatic soil $CO_2$ efflux measurement system with a portable chamber (LI-COR Inc., Lincoln, NE, USA). The ST and moisture content at a depth of five cm adjacent to the collar were simultaneously measured using a LI-COR 8150-203 thermocouple probe and a LI-COR 8150-204 $ECH_2O$ Model EC-5 soil volumetric moisture content probe, respectively (Decagon Device, Inc., Pullman, WA, USA).

### Plant aboveground and root biomass

Aboveground biomass was measured at the end of December 2014 and during May, July, and September 2015; root biomass was measured during May, July, and September 2015. Aboveground biomass was removed from a randomly located $0.5 \times 0.5$ m quadrant in each plot and oven-dried together with litter at 65 °C to a constant weight. Root biomass was collected using an eight cm diameter soil drill sampler, and three cores were collected at 0–20 cm deep from random locations in each plot to be mixed as one composite sample. The roots were washed and oven-dried at 65 °C to a constant weight.

### Soil microbial biomass and soil nutrients

The collected soil samples were separated from the root biomass. Part of each sample was passed through a two-mm sieve to analyse soil microbial biomass carbon (MBC), dissolved organic carbon (DOC), and phospholipid fatty acids (PLFAs). The composition of the microbial community composition in each fraction was assessed using PLFA analysis following the method of *Bossio & Scow (1998)*. MBC was estimated using the chloroform fumigation-extraction method and calculated as the difference in extracted organic carbon between the fumigated and control parts divided of the soil sample divided by a conversion factor of 0.45 (*Vance, Brookes & Jenkinson, 1987*). DOC was determined as the extracted organic carbon in the control soil (*Vance, Brookes & Jenkinson, 1987*). PLFAs (including i14:0, 15:0, i15:0, a15:0, 16:0, 10me16:0, i16:0, 16:1w5c, 16:1w7c, 10me17:0, a17:0,

i17:0, cy17:0, 17:1w8c, 18:0, 10me18:0, 18:1w5c, 18:1w7, 18:1w9c, 18:2w6c cy19:0, 20:1w9c, and 20:4w6c) were identified by bacterial biomarkers, while fungal PLFAs, actinobacterial PLFAs and protozoal PLFAs were assessed following the method of *Huang et al. (2014)*.

The balance of each soil sample was passed through a 0.25-mm sieve for analysis of soil TN and total phosphorus (TP). TN and TP were measured by the semi-micro-Kjeldahl method (*Bremner, 1960*) with a PerkinElmer 2400 series II CHNS/O analyzer (PerkinElmer, Inc., Norwalk CA, USA) and a SPECTRO ARCOS EOP analyzer (Spectro, Inc., GER), respectively.

## Data analysis

$Q_{10}$ is defined as the rate of increase in Rs as a consequence of increasing the temperature by 10 °C and is derived from the following relationships:

$$Rs = a \cdot e^{bT} \tag{1}$$

$$Q_{10} = e^{10b} \tag{2}$$

where Rs ($\mu mol \cdot m^{-2} \cdot s^{-1}$) is Rs, $T$ (°C) is the ST at a depth of five cm, and $a$ and $b$ are fitted parameters.

The response of Rs to SM was modelled using a quadratic polynomial function of the following form:

$$Rs = ex^2 + fx + g \tag{3}$$

where Rs ($\mu mol \cdot m^{-2} \cdot s^{-1}$) is Rs; $x$ (%) is the SM content at a depth of five cm; and $e$, $f$, and $g$ are fitted parameters.

All statistical analyses were performed using SPSS 12.0 (SPSS for Windows, Chicago, IL, USA), and the structural equation model (SEM) analyses were conducted using SPSS AMOS 24.0.0 (SPSS Inc., Chicago, IL, USA).

## RESULTS

### Soil temperature and moisture

Soil temperature and SM varied significantly among the three grazing systems in the same month during the period of observation ($P < 0.05$, Figs. 1A and 1B). STs exhibited consistent seasonal dynamics across all grazing systems, with the maximum ST occurring in July 2015 and the minimum occurring in December 2014 (Fig. 1A). The low-vegetation cover plots (year-long grazing and rest-rotation grazing systems) had higher STs than the high-vegetation cover plots (grazing exclusion system) due to the absence of canopy shading.

The SM content fluctuated substantially in response to irregular rainfall, peaking in September 2015 (Fig. 1B). The average SM content during observation was 7.89%, 9.87%, and 10.89% for the year-long grazing, rest-rotation grazing, and grazing exclusion plots, respectively.

### Soil respiration and $Q_{10}$

Soil respiration showed significant variation among the grazing systems in all months ($P < 0.05$). The Rs ranges were 0.07–1.89, 0.06–2.57, and 0.03–3.47 $\mu mol \cdot m^{-2} \cdot s^{-1}$ for the

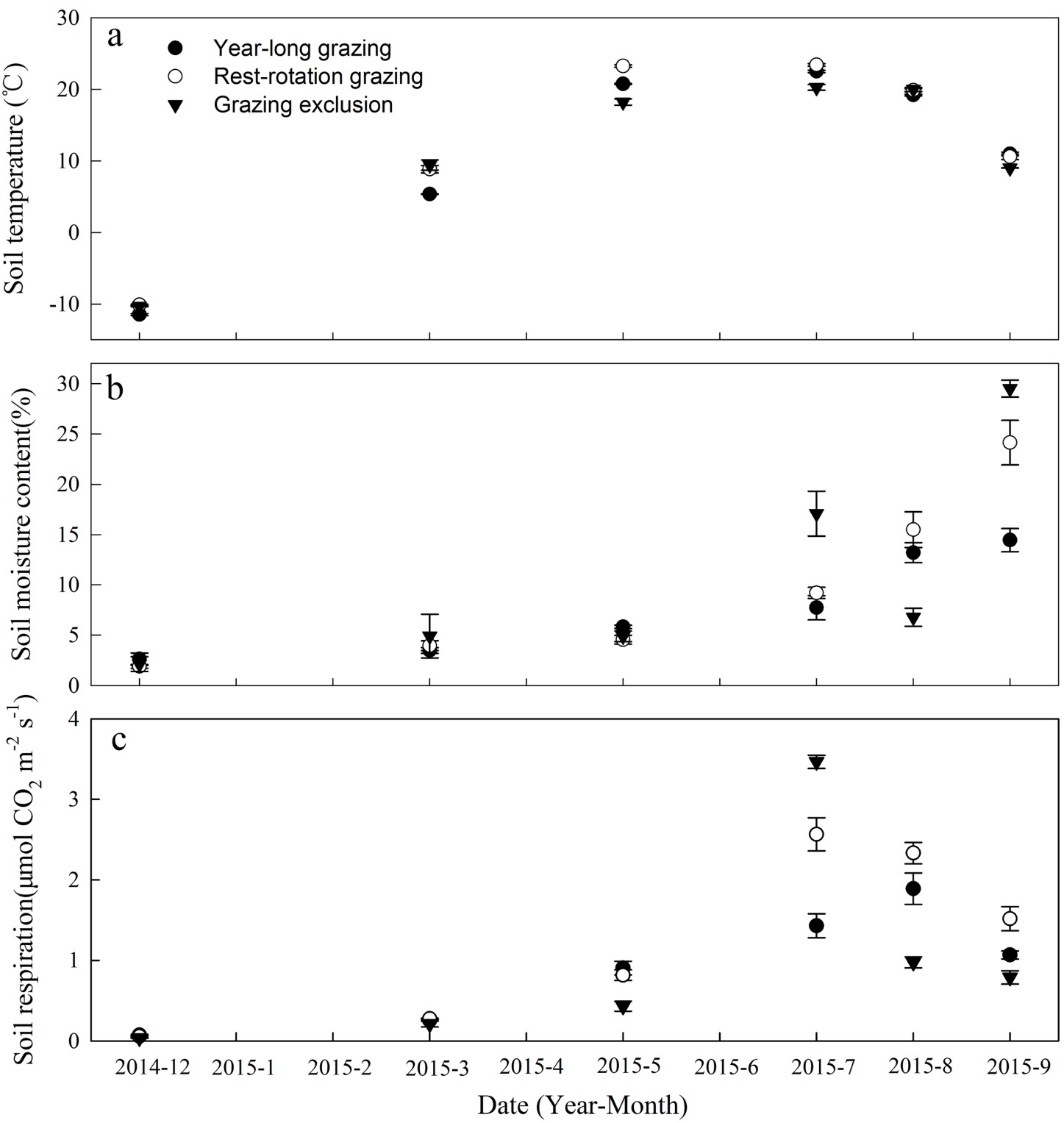

**Figure 1** Dynamics of (A) soil temperature, (B) soil moisture content, and (C) soil respiration under three different grazing systems from December 2014 to September 2015.

**Table 1 Temporal variation in the soil respiration–temperature relationship (Eq. (1)) and $Q_{10}$ among three grazing systems. The $R^2$ and $P$-values refer to Eq. (1).**

| Date Year/month | Grazing systems | Equation (1) $Rs = a \cdot e^{bT}$ | $Q_{10}$ | $R^2$ | $P$ |
|---|---|---|---|---|---|
| 2014-12 | Year-long grazing | $2.298e^{0.308T}$ | 21.76 | 0.804 | <0.001 |
| | Rest-rotation grazing | $1.191e^{0.331T}$ | 27.39 | 0.740 | <0.001 |
| | Grazing exclusion | $0.541e^{0.298T}$ | 19.69 | 0.644 | <0.001 |
| 2015-03 | Year-long grazing | $0.14e^{0.100T}$ | 2.72 | 0.506 | <0.001 |
| | Rest-rotation grazing | $0.113e^{0.092T}$ | 2.51 | 0.655 | <0.001 |
| | Grazing exclusion | $0.111e^{0.06T}$ | 1.82 | 0.677 | <0.001 |
| 2015-05 | Year-long grazing | $0.171e^{0.067T}$ | 1.95 | 0.736 | <0.001 |
| | Rest-rotation grazing | $0.288e^{0.053T}$ | 1.70 | 0.883 | <0.001 |
| | Grazing exclusion | $0.166e^{0.049T}$ | 1.63 | 0.889 | <0.001 |
| 2015-07 | Year-long grazing | $0.810e^{0.024T}$ | 1.28 | 0.743 | <0.001 |
| | Rest-rotation grazing | $0.975e^{0.041T}$ | 1.51 | 0.883 | <0.001 |
| | Grazing exclusion | $1.496e^{0.04T}$ | 1.49 | 0.750 | <0.001 |
| 2015-08 | Year-long grazing | $0.614e^{0.056T}$ | 1.75 | 0.920 | <0.001 |
| | Rest-rotation grazing | $0.950e^{0.044T}$ | 1.55 | 0.901 | <0.001 |
| | Grazing exclusion | $0.545e^{0.028T}$ | 1.32 | 0.789 | <0.001 |
| 2015-09 | Year-long grazing | $0.655e^{0.043T}$ | 1.54 | 0.746 | <0.001 |
| | Rest-rotation grazing | $0.948e^{0.043T}$ | 1.54 | 0.915 | <0.001 |
| | Grazing exclusion | $0.125e^{0.193T}$ | 6.89 | 0.855 | <0.001 |

year-long grazing, rest-rotation grazing and grazing exclusion systems, respectively (Fig. 1C). Rs was lowest in December 2014 for all three grazing systems. It peaked in July for the grazing exclusion and rest-rotation grazing plots and in August for the year-long grazing plots (Fig. 1C). Averaged across the period of observation, Rs was highest in the rest-rotation plots (1.26 μmol·m$^{-2}$·s$^{-1}$), followed by the grazing exclusion (0.98 μmol·m$^{-2}$·s$^{-1}$) and year-long grazing (0.94 μmol·m$^{-2}$·s$^{-1}$) plots. During the 2015 growing season (from May to September), the average Rs values were 1.32, 1.81, and 1.42 μmol·m$^{-2}$·s$^{-1}$ for the year-long grazing, rest-rotation grazing, and grazing exclusion systems, respectively.

Significant exponential relationships were found between Rs and ST for all grazing systems and sample times, with $R^2$ values ranging from 0.506 to 0.920 (Table 1). The $Q_{10}$ values for the grazing exclusion plots were lower than those for the year-long grazing plots, except in July and September 2015, and were lower for the rest-rotation plots than for the year-long grazing plots, except in December 2014, July 2015, and September 2015 (Table 1). Very high $Q_{10}$ values were recorded for all grazing systems in December 2014, and a relatively high value was recorded for the grazing exclusion system in September 2015, indicating the heightened sensitivity of Rs to temperature at these times (Table 1).

A quadratic relationship between Rs and SM was derived from the monthly SM measurements in each grazing system (Fig. 2). All relationships were significant ($P < 0.001$), with $R^2$ values ranging from 0.69 to 0.77. The effect of increasing SM on Rs was non-linear at this study site.

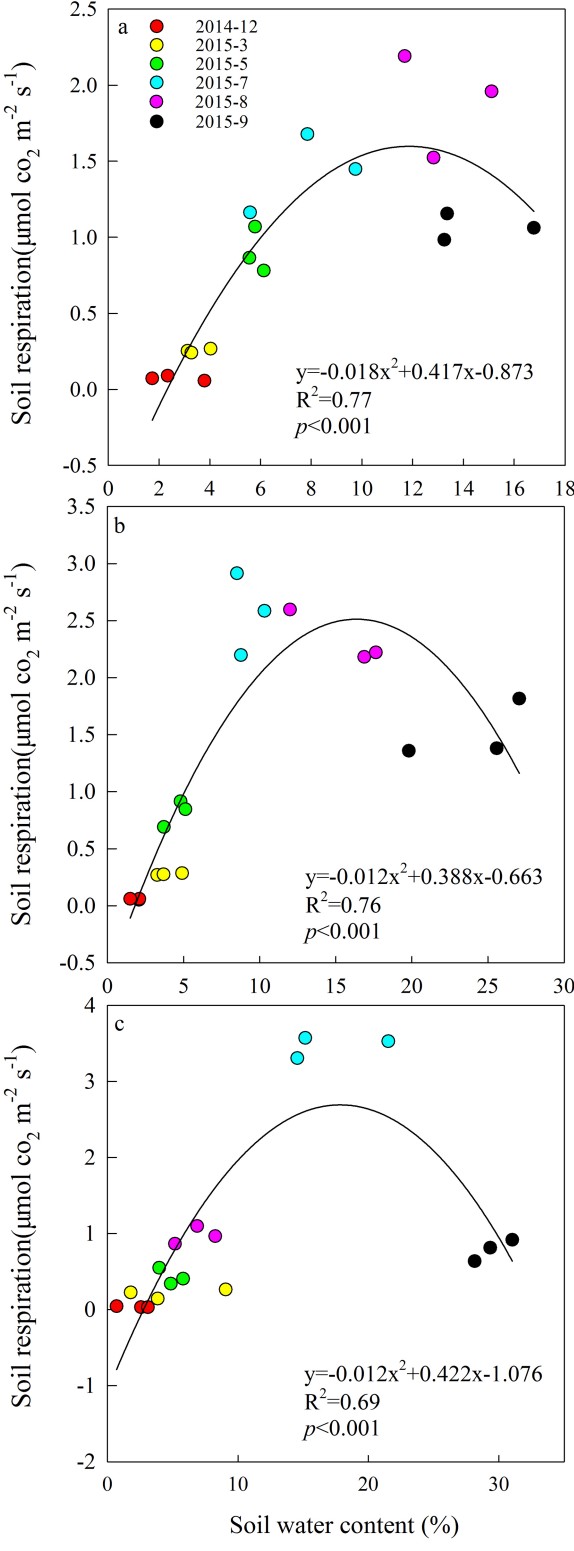

**Figure 2 Quadratic relationships between soil respiration and soil moisture from December 2014 to September 2015 in (A) year-long grazing system, (B) rest-rotation grazing system, (C) grazing exclusion system.** $R^2$ is the goodness of fit for the function, and the $P$-value is the significance of the overall function.

## Plant and soil properties

Aboveground biomass under the three grazing system increased over time within a growing season, and it was higher in the grazing exclusion system than in the other two systems for the same month (Fig. 3A). Root biomass showed remarkable variation between months, with the maximum biomass occurring in July 2015 (Fig. 3B). Root biomass was significantly higher in the rest-rotation grazing system than in the year-long grazing system in July and September and higher than in the grazing exclusion system in May, July, and September ($P < 0.05$). Rs increased linearly with root biomass in all grazing systems as the growing season progressed (Fig. 4).

Total phosphorus and TN remained steady during the experiment in all grazing systems (Figs. 3C and 3D). The average TP content was 0.37, 0.32, and 0.42 g kg$^{-1}$, while the average TN content was 1.57, 1.63, and 2.36 kg$^{-1}$ for the year-long grazing, rest-rotation grazing, and grazing exclusion systems, respectively. Both the TN and TP contents of the grazing exclusion system were significantly higher than the other two systems ($P < 0.05$, Figs. 3C and 3D). MBC increased during the 2015 growing season in all systems (Fig. 3E). MBC of the grazing exclusion system was 49% higher than that of the year-long grazing system in July and 42% higher in September. DOC decreased as the growing season progressed and was higher in the year-long grazing system than in the other two systems (Fig. 3F).

## Phospholipid fatty acids

The total PLFAs declined at first and then increased from December 2014 to September 2015, and each kind of PLFA increased during the course of the growing season (Figs. 5A–5E). The mean total PLFA values for the year-long grazing, rest-rotation grazing, and grazing exclusion systems in the growing season were 29.77, 31.9, 35.88 nmol g$^{-1}$, respectively. The highest levels for all types of PLFAs were observed in the grazing exclusion system during September (Figs. 5A–5E). The average ratios of fungi to bacteria during the growing season were 0.17, 0.17, and 0.14 in the year-long grazing, rest-rotation grazing, and grazing exclusion systems, respectively.

## Relationships among $Q_{10}$, Rs, and other factors under different grazing systems

Soil respiration and $Q_{10}$ were affected by grazing, climate, DOC, plant, and soil microbes; these factors were related to each other. Therefore, the SEM was used to explore the influence mechanism of Rs and $Q_{10}$. The SEM explained 70% and 63% of the variance of Rs and $Q_{10}$, respectively, under different grazing systems (Fig. 6). Among the variables influencing Rs and $Q_{10}$, the increase in PLFAs and ST resulted in a highly significant enhancement of Rs ($P < 0.001$). The standardized total effect coefficients of PLFAs and ST on Rs were 0.634 and 0.621, respectively. The increase in the ST by seasonal variation resulted in a highly significant decrease in $Q_{10}$ ($P < 0.001$), and the standardized total effect coefficient was −0.775. Grazing had an insignificant negative direct effect (−0.051) on Rs, and an indirect negative effect (−0.078) by decreasing aboveground biomass, PLFAs, and SM. Grazing had a small indirect positive effect (0.026) by decreasing the aboveground

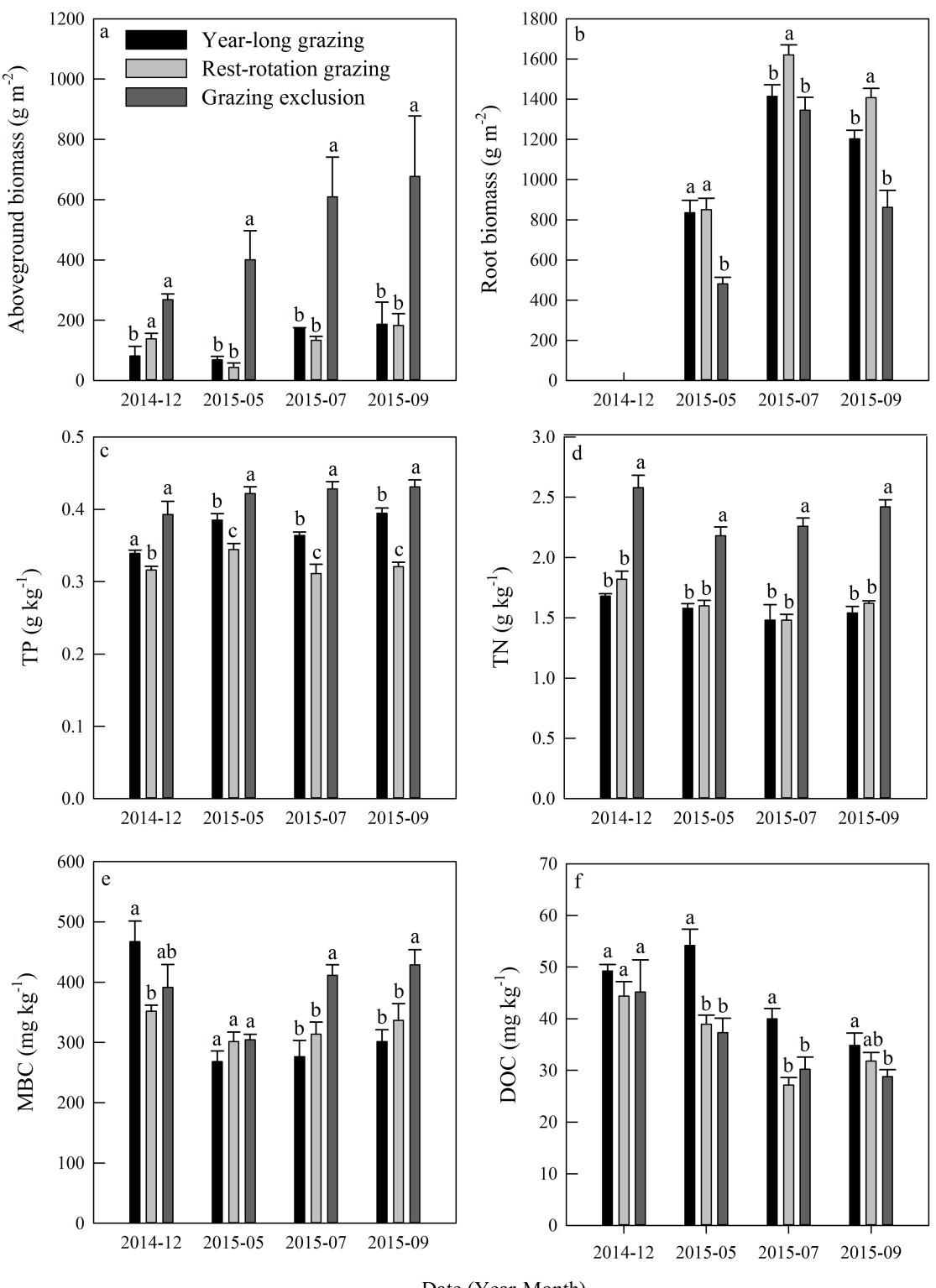

**Figure 3** Dynamics of (A) aboveground biomass, (B) root biomass, (C) total phosphorus (TP), (D) total nitrogen (TN), (E) microbial biomass carbon (MBC), and (F) dissolved organic carbon (DOC) under three different grazing systems from December 2014 to September 2015.

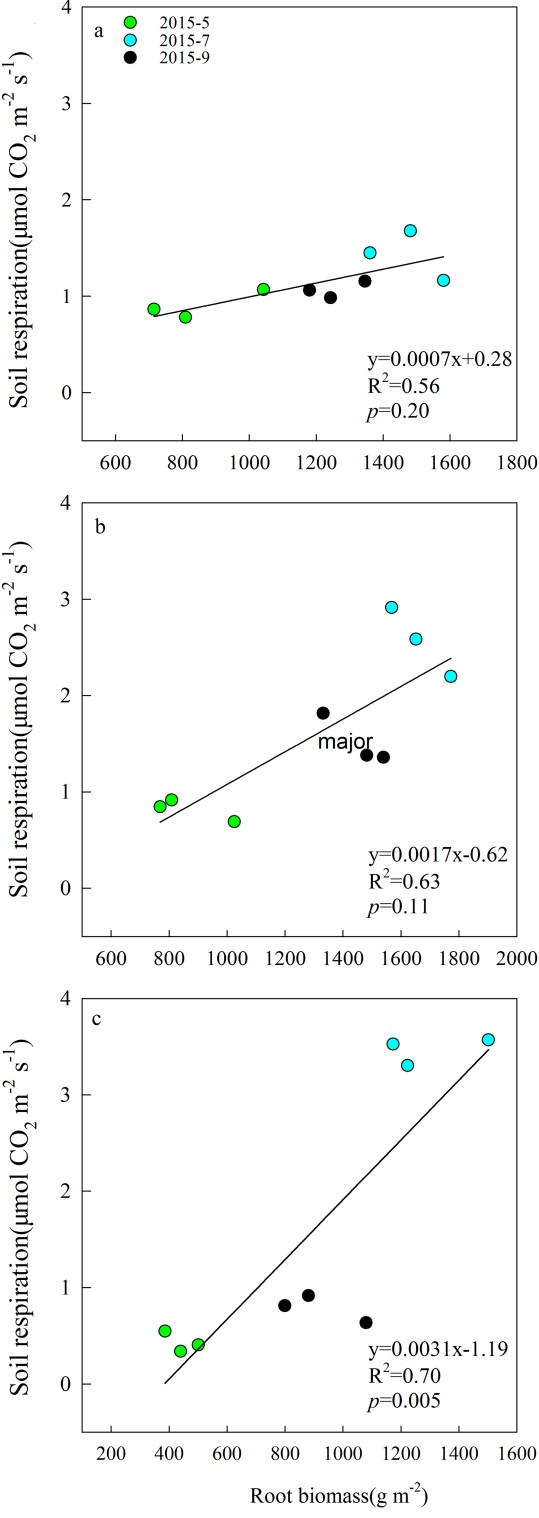

**Figure 4 Linear relationships between soil respiration and root biomass in May, July, and September 2015 under (A) year-long grazing system, (B) rest-rotation grazing system, and (C) grazing exclusion system.** $R^2$ is the goodness of fit for the function, and the $P$-value is the significance of the overall function.

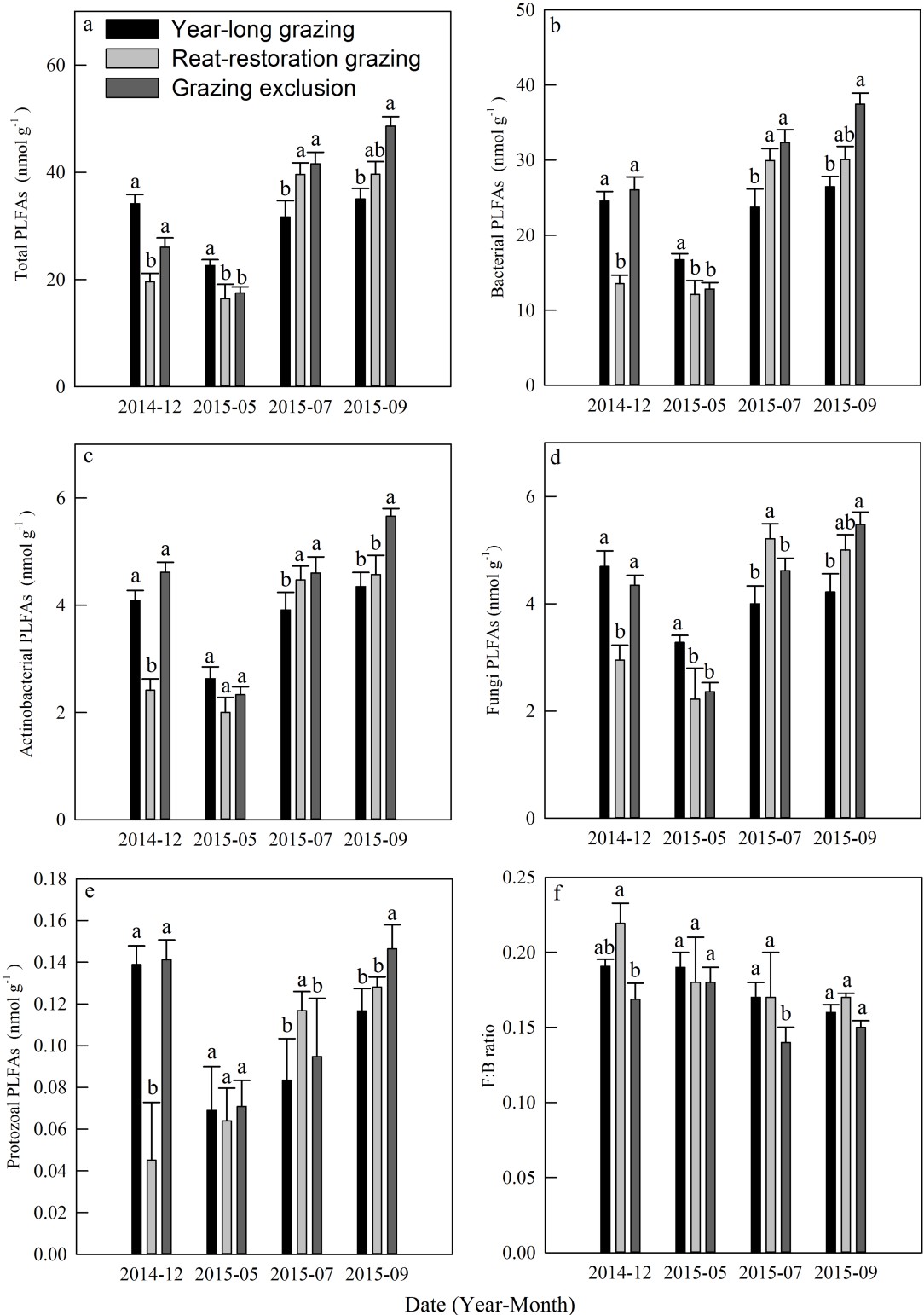

**Figure 5 Dynamics of phospholipid fatty acids (PLFAs, nmol g$^{-1}$) under different grazing systems from December 2014 to September 2015.** (A) Total PLFAs, (B) bacterial PLFAs, (C) actinobacterial PLFAs, (D) fungal PLFAs, (E) protozoal PLFAs, and (F) the ratio of fungal: bacterial PLFAs (F:B ratio). The different letters represent significant differences ($P < 0.05$) among the different grazing systems in the same month. The error bars represent the standard errors.

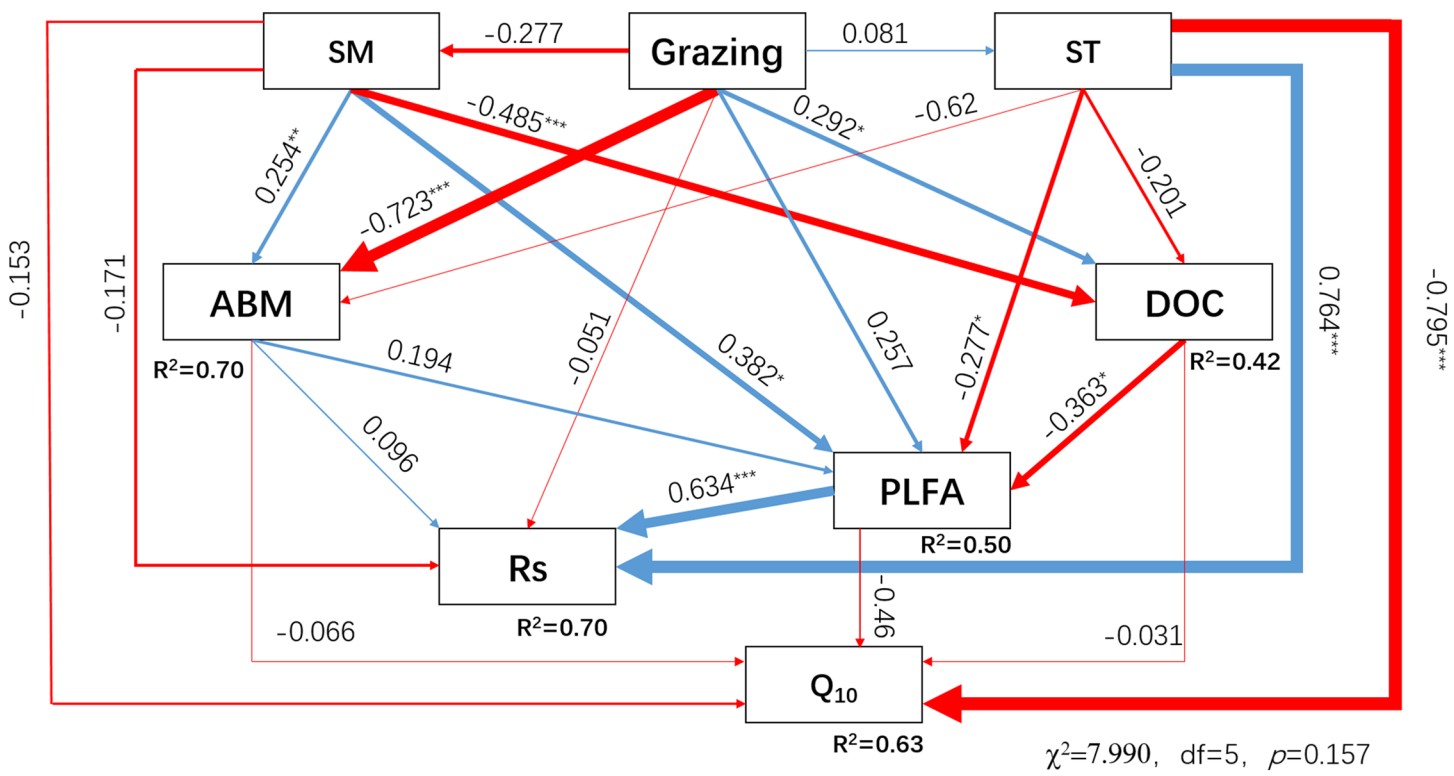

**Figure 6 Structural equation model (SEM) describing the effects of grazing, environmental factors, and biotic factors on soil respiration and $Q_{10}$.** The SEM was performed with data from December 2014 and from May, July, and September 2015. ST, SM, ABM, DOC, PLFA, Rs, and $Q_{10}$ represent the soil temperature, soil moisture, aboveground biomass, dissolved organic carbon, phospholipid fatty acids, soil respiration, and temperature sensitivity for Rs, respectively. The blue or red lines, the direction of the arrows and the thickness of the lines indicate positive or negative effects, the direction of the effects and the severity of the effects, respectively. $R^2$ denotes the proportion of variables explained by these drivers. The numbers close to the lines are the standardized direct influence coefficients. The *, **, and *** at the top right of the numbers indicate that the effects were significant at $P < 0.05$, $P < 0.01$, and $P < 0.001$, respectively. $\chi^2$, chi-square; d*f*, degrees of freedom; *P*, probability level.

**Table 2 Standardized total effects of ST, SM, grazing, ABM, DOC, and PLFAs on ABM, DOC, Rs, and $Q_{10}$.**

|  | Grazing | SM | ST | ABM | DOC | PLFA |
|---|---|---|---|---|---|---|
| SM | −0.277 |  |  |  |  |  |
| ST | 0.081 |  |  |  |  |  |
| ABM | −0.799 | 0.254 | −0.062 |  |  |  |
| DOC | 0.409 | −0.485 | −0.201 |  |  |  |
| PLFA | −0.174 | 0.607 | −0.216 | 0.194 | −0.363 |  |
| $Q_{10}$ | 0.026 | −0.183 | −0.775 | −0.075 | −0.014 | −0.046 |
| Rs | −0.129 | 0.238 | 0.621 | 0.219 | −0.230 | 0.634 |

Note:
ST, SM, ABM, DOC, PLFAs, Rs, and $Q_{10}$ represent soil temperature, soil moisture, aboveground biomass, dissolved organic carbon, phospholipid fatty acids, soil respiration, and temperature sensitivity of soil respiration, respectively.

biomass and increasing SM on $Q_{10}$. The result of the SEM indicated that increased grazing inhibited Rs and enhanced $Q_{10}$. When the standardized total effects were compared (Table 2), the effect of grazing on Rs and $Q_{10}$ was lower than the effects of ST or moisture.

## DISCUSSION

### Response of plant biomass, soil properties, and microbial community to grazing

Grazing is well-acknowledged to have impacts on the growth of vegetation (*Porensky et al., 2016*), basic soil properties, and soil microbial biomass (*Raich & Nadelhoffer, 1989*) in grassland ecosystems. In the present study, we found that root biomass in year-long grazing and rest-rotation grazing systems were higher than that in grazing exclusion systems (Fig. 3). These results can be explained by the grazing optimization hypothesis proposed by *McNaughton (1979)*, which suggested that moderate grazing stimulates vegetation regeneration. Moreover, plants would supply more nutrients to roots instead of shoots as a change in growth strategy due to grazing (*Bai et al., 2015*; *Frank, Kuns & Guido, 2002*). But the root biomass increased significantly only in rest-rotation grazing systems, it indicating that rest-rotation grazing could result in economic benefits and grassland restoration.

We found that soil total carbon and TP were more abundant in the grazing exclusion system than in the year-long grazing and rest-rotation grazing systems (Fig. 3). Similarly, previous studies also found that grazing had a negative effect on the storage of soil nutrients (*Han et al., 2008*), whereas fencing improved the physical and chemical properties of soil (*Dormaar, Smoliak & Willms, 1989*). Loss of soil nutrient elements may limit the growth of vegetation and the soil microbial community, and it also could be an important reference index for the establishment of grassland management mode. In contrast to inorganic elements, DOC increased with intensified grazing during the growing season, except in July of 2015 when the rest-rotation grazing system was being implemented (Fig. 3F). DOC has been recognized to play an important role in regulating soil carbon mineralization and Rs in grassland ecosystems (*Zhou et al., 2016*). The increase in DOC caused by grazing provided more respiration substrates, which may consequently enhance Rs.

Soil microbes are crucial in regulating Rs by transforming soil carbon into $CO_2$ flux during decomposition (*Chapin, Matson & Vitousek, 2011*). We found that grazing generally suppressed the MBC in the 2015 growing season (Fig. 3E), which is consistent with results from *Bardgett & Leemans (1995)*. The result of the PLFA assays also showed that grazing significantly decreased the total PLFAs, bacterial PLFAs and actinomycelial PLFAs in the middle and late growing season (Fig. 5). The decrease in microbial biomass is likely attributed to livestock trampling disturbing the soil microenvironment and thus inhibiting soil microbe reproduction, although the increase in the root biomass caused by grazing may provide more exudates, promoting microbial growth. We also found that the F:B ratio in year-long grazing and rest-rotation grazing systems was higher than that in grazing exclusion systems. This increased F:B ratio by grazing might be attributed to the grazing-induced increase in soil C:N ratio by increasing SOC and decreasing TN since substrates with high C:N ratios are conducive to fungal growth. Fungi play an important role in a large number of soil ecosystem processes, particularly in nutrient cycling and maintaining soil structure (*Wal et al., 2006*). As a higher F:B ratio in soil could reflect a more

stable ecosystem status (*Vries et al., 2006*), moderate grazing might to some extent improve the stability of grassland ecosystems.

## Relationships between soil respiration and other factors under grazing

In accordance with the results of previous studies (*Bao et al., 2016*; *Li & Sun, 2011*), tight linkages among Rs and plant and soil properties were found in this study (Fig. 6). Compared to other factors, ST had a significantly direct effect on Rs ($P < 0.001$) in the constructed SEM. Rs in all grazing systems increased exponentially with rising ST, which is consistent with other findings (*Curiel Yuste et al., 2004*; *Zhang et al., 2015b*). Nonetheless, we also found that in August 2015, Rs decreased by 71.8% compared to that in July in the grazing exclusion system, while ST was similar between these two months. This result indicates that in addition to ST, other factors, for example, SM, might also contribute to monthly variation in Rs.

Soil moisture is one of the factors limiting ecological recovery in semiarid regions (*Zhang et al., 2015b*). SM was considered a dominant driver of Rs when temperature was above 20 °C in a semiarid steppe ecosystem in Spain (*Rey et al., 2011*). According to the results of the SEM, SM directly inhibited Rs (Fig. 6). It is suggested that high SM may reduce Rs because the water in the soil interstices affects gaseous diffusion (*Liu et al., 2014*). However, high SM can also enhance the transport of organic matter among soil and metabolic activity of soil microbes (*Michalzik et al., 2003*). The positive and negative effects of moisture on Rs exist simultaneously, and their relationship fit the quadratic function well, echoing the findings of *Suseela et al. (2012)*. It may also explain why the Rs of the three grazing systems peaked in different months. In July and August, higher ST caused moisture to become a major limiting factor of Rs; therefore, the difference in Rs between year-long grazing and grazing exclusion aligned with changes in SM. This resulted in the Rs of grazing exclusion systems peaking in July, while the Rs of year-long grazing systems peaked in August. In the rest-rotation grazing system, Rs peaked in July but did not show a significant difference ($P < 0.05$) between July and August. This finding was probably due to root biomass being the highest in rest-rotation grazing systems (Fig. 3), so rhizosphere respiration (Rr) became the main component of Rs, and it was thus less sensitive to surface SM. Due to the non-linear effect of SM on Rs, identifying the non-linearities and thresholds could reduce some uncertainties associated with climate change decision-making.

In the present study, the average Rs from May to September was highest in the rest-rotation grazing system (1.81 $\mu mol \cdot m^{-2} \cdot s^{-1}$), followed by the grazing exclusion system (1.41 $\mu mol \cdot m^{-2} \cdot s^{-1}$) and the year-long grazing system (1.32 $\mu mol \cdot m^{-2} \cdot s^{-1}$). This result indicated that grazing had a significant impact on Rs in a temperate typical steppe that was not only inhibitory, thus contradicting our first hypothesis that grazing inhibited Rs. Similarly, a study conducted under different land uses in Inner Mongolia reported that the average Rs during the growing season was 1.43 ± 0.04 $\mu mol \cdot m^{-2} \cdot s^{-1}$ for free grazing (only during the growing season with nine sheep $ha^{-1}$) and 1.37 ± 0.03 $\mu mol \cdot m^{-2} \cdot s^{-1}$ for enclosed grassland grazing (*Xue & Tang, 2018*). In contrast to the findings of the study, the higher Rs observed under rest-rotation grazing system in our study was likely due to

the lower grazing intensity (2–4.5 vs 9 sheep ha$^{-1}$). Rest-rotation grazing accelerated the soil carbon mineralization rate, and also reduced soil carbon stocks and increased the greenhouse effect.

In the SEM, grazing showed a minor negative direct effect on Rs, which could be explained as an increase in soil bulk density and compactness by grazing that might block gas exchange. The indirect effects of grazing on Rs were mainly mediated by plant and microbial factors. Our study found that the mean Rs and root biomass were highest under rest-rotation grazing, probably because grazing accelerated the nutrient cycle (*Allard et al., 2007*) and increased the transfer of plant biomass to roots (*Milchunas & Lauenroth, 1993*). Previous research (*Tu et al., 2013*; *Sun et al., 2013*) has shown that environmental factors affect root respiration through changes in root biomass. The root biomass was higher in grazing systems, and Rs increased linearly with root biomass in all grazing systems (Fig. 4). This finding suggests that grazing enhanced Rs by increasing root biomass.

Soil microbial respiration was one of the main contributors to Rs. A previous study showed that the response of Rs to grazing was positively correlated with soil microbial biomass under moderate and heavy grazing intensities (*Zhao et al., 2017*). In addition, the substrate supply from photosynthesis is a main factor affecting Rs, and the aboveground biomass is proportional to Rs (*Kuzyakov & Gavrichkova, 2010*). In this study, Rs was positively correlated positively to aboveground biomass and microbial biomass (Fig. 6). Previous studies have shown that DOC was the substrate of soil microbial respiration, and its content and degree of decomposition positively correlated with Rs (*Moinet et al., 2016*). However, DOC had a negative effect on Rs in this study mainly because the decrease in soil microbes by grazing weakened the restrictive effect of DOC on Rs. In conclusion, grazing inhibited respiration by suppressing aboveground biomass, SM and soil microbial biomass. These results were consistent with the second hypothesis that grazing enhanced Rr but inhibited microbial respiration.

## The relationships between the temperature sensitivity of soil respiration and other factors under grazing

Previous studies have reported that the $Q_{10}$ of Rs is negatively correlated with ST in northern temperate grasslands (*Flanagan & Johnson, 2005*). The higher values of $Q_{10}$ in December of 2014 and March of 2015 in our study were consistent with that result, indicating that Rs could be extremely sensitive to temperature changes at very low STs. The decrease in the $Q_{10}$ with increasing temperature could be related to a decline in $Q_{10}$ of microbial respiration, which contributed a great proportion to Rs. A previous study conducted in forest soils found that the $Q_{10}$ of SOM decomposition rates decreases with the increase of ST and moisture (*Schindlbacher et al., 2010*), which was similar to the $Q_{10}$ of Rs in our study (Fig. 6). This finding can be explained by the differential response of enzyme activity and substrate affinity to increased temperature. Enzyme activity was inhibited at low temperatures and could be enhanced by ST, but the decline in enzyme substrate affinity with increased temperature could offset the promotion of soil enzyme activity (*Li et al., 2017*), resulting in higher $Q_{10}$ and decomposition potential of SOM

at low temperatures. Rs increased rapidly after removing the temperature limit; the limiting effect of temperature and the rate of increase in Rs decreased gradually in this process, meaning that $Q_{10}$ decreased with increasing ST. $Q_{10}$ decreased more slowly as limitations were removed.

The main components of Rs are autotrophic respiration (Rr) and heterotrophic respiration (microbial respiration). Sensitivity of different components to temperature determine the total $Q_{10}$ of Rs. Evidence regarding the response of $Q_{10}$ to grazing has been inconsistent. $Q_{10}$ increased under grazing exclusion in a meadow grassland (*Chen et al., 2016*) but was stimulated by grazing in another temperate grassland (*Paz-Ferreiro et al., 2012*). In this study, the temperature sensitivity of Rs in the two grazing systems was higher than that in the grazing exclusion system. Consistent with the third hypothesis, the increased $Q_{10}$ from grazing in this study could be explained by the higher fine root biomass in grazing systems and higher $Q_{10}$ value for Rr (*Boone et al., 1998*). In addition, the "rhizosphere priming effect" increased the availability of labile carbon by root exudation or fresh litter, which enhanced the $Q_{10}$ of heterotrophic respiration (*Hopkins et al., 2013*; *Zhu & Cheng, 2011*). These results indicate that Rs in the grassland is more susceptible to rising temperature under grazing.

Compared with year-long grazing, rest-rotation grazing could be a reasonable avenue to grassland sustainable development, but the increased of soil $CO_2$ emissions and loss of soil nutrient elements caused by rest-rotation grazing are inescapable. Given grassland degradation caused by long-term grazing in typical steppe, grazing exclusion could be an effective way to promote grassland restoration. Due to lack of long-term observational data and multiple replicates in each plot, and because respiration of different components cannot be characterized accurately by microbial biomass and root biomass, the results of this study have some limitations. Future studies can set multiple gradients of grazing time or intensity to explore the varying effects of grazing, and focus on changes of different components of Rs under grazing by more effective means. Long-term measurements of Rs and soil organic matter should be performed.

## CONCLUSIONS

In this study, the mean Rs varied significantly among different grazing systems. Diverse effects of grazing management on soil $CO_2$ flux occurred in this temperate grassland, as the mean Rs was lowest in the year-long grazing system but highest in the rest-rotation grazing system. The difference in Rs among the grazing systems was closely associated with changes in ST, SM, potential substrate availability and soil microbial activity. Since rhizospheric respiration increased with increasing root biomass stimulated by grazing and was more sensitive to increased temperature, the temperature sensitivity ($Q_{10}$) of Rs was enhanced in grazing systems. Therefore, grazing has the potential to make Rs more susceptible to global warming. In addition, the occurrence of the highest contents of TN, TP, and MBC in the grazing exclusion system demonstrated the value of grazing exclusion for conserving soil nutrients and microbial biomass. Higher F:B ratios in the grazing systems were likely due to more suitable soil management. By altering plant roots and their control over the soil microbial community, grazing profoundly influences soil

decomposition and thus can regulate the carbon budget in grassland ecosystems. A better understanding of the effects of grazing on Rs has practical implications for ecological restoration in a typical steppe.

### Funding
This work was only supported by the National Natural Science Foundation of China (31770519) and the National Key R&D Program of China (2017YFC0503805). There was no additional external funding received for this study. The funders had no role in study design, data collection and analysis, decision to publish, or preparation of the manuscript.

### Grant Disclosures
The following grant information was disclosed by the authors:
National Natural Science Foundation of China: 31770519.
National Key R&D Program of China: 2017YFC0503805.

### Competing Interests
The authors declare that they have no competing interests.

### Author Contributions
- Cheng Nie conceived and designed the experiments, performed the experiments, analyzed the data, prepared figures and/or tables, authored or reviewed drafts of the paper, approved the final draft.
- Yue Li conceived and designed the experiments, performed the experiments, analyzed the data.
- Lei Niu conceived and designed the experiments, performed the experiments, analyzed the data.
- Yinghui Liu conceived and designed the experiments, contributed reagents/materials/analysis tools, authored or reviewed drafts of the paper, approved the final draft.
- Rui Shao authored or reviewed drafts of the paper.
- Xia Xu contributed reagents/materials/analysis tools.
- Yuqiang Tian contributed reagents/materials/analysis tools.

### Field Study Permissions
The following information was supplied relating to field study approvals (i.e., approving body and any reference numbers):
Field experiments were approved by the Duolun Restoration Ecology Research Station (part of the Institute of Botany, Chinese Academy of Sciences).

### Data Availability
The raw data of soil respiration, PLFAs and soil properties are available in the Supplemental Files.

## Supplemental Information

Supplemental information for this article can be found online at http://dx.doi.org/10.7717/peerj.7112#supplemental-information.

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
