# Peer review of "Soil respiration and its Q10 response to various grazing systems of a typical steppe in Inner Mongolia, China"

_PeerJ, doi:10.7717/peerj.7112_

## Round 0.1 · original submission · Major Revisions

Dear Authors,

Your manuscript has been reviewed by three external reviewers and me. I agree with most comments that there is interesting data presented in this manuscript. In addition to the comments from expert reviewers, I have two major issues that will need to be addressed that I think may increase readability and readership of the manuscript.

1- The novelty of the research and/or major outcomes of the study are not clearly presented in the manuscript. This is due to the ineffective writing style at times; the lack of description of the methods and the poorly written hypotheses. Rather, hypotheses are written as predictions which may not be always falsifiable. Besides, answering the predictions posed by authors do not inform about the system, mechanisms, or implications even though the data analyses makes fair attempts at doing so.This is because the

2- I am a bit puzzled by the use of references. Most of them are from their work or the system, which limits readability and extrapolation of results. There are no references on the limits of interpretation of seasonal Q10 values, autotrophic and heterotrophic components of soil respiration and the role of photosynthesis in influencing rates of soil respiration and their components. In Hopkins et al., 2013 for instance we critically reviewed the temperature-soil respiration relationships and there is little mention of those issues in here. Broadening the citations and placing results in the context of the burning questions of the soil respiration community can help bringing their results into the larger community.

Reviewers bring specific comments that I think will greatly improve the manuscript.

Reviewer 1 ·

Basic reporting

The aim of the paper is not clear with poor language expression.More similar reports have been published.This paper has no new ideas.More recent references were not cited.

Experimental design

The design is not good ,and the experimental tome is too short.The obtained data are not enough to get the conclusions.

Validity of the findings

No interacting discussions were provided.More work is repeated.

Additional comments

Your paper is more about data collection,and there is no new ideas and new integrated conclusions supported by current data set.More similar reports have been published.Only limited references were cited.

Reviewer 2 ·

Basic reporting

Nie et al conducted an experiment in a temperate grassland ecosystem in Inner Mongolia to study the effects of grazing on soil respiration and its temperature sensitivity. The manuscript is well organized and fits to the scope of the Journal. The results showed that different grazing treatments showed different soil respiration and its Q10, and soil microclimate and root biomass are the main factors determining RS and Q10 under various grazing treatments. The results are generally well presented as tables or figures and the writing is easy to follow.

Unfortunately, the methods and discussion need further improvement. The methods should be more detailed and the discussion need to be more in-depth and developed, instead of simply comparing with previous findings.

L32 Basic soil properties are too broad. The specific soil properties affected by grazing in this study may be more appropriate.

Experimental design

The authors provided three hypotheses to be verified. The hypotheses should be more specific, e.g. how grazing systems would change soil respiration and temperature and by what mechanisms.

Why do you conduct measurement in Dec.? If you want to explore the dynamics of RS in winter, one measurement is not enough (L100).

Also, the measurement period is too short (less than a year), which is the main limitation of this study.

Only one collar may be not enough to overcome the influence of spatial variation of soil respiration, as your study plot is 50 m2 (L101).

Do you measure soil temperature and soil moisture adjacent to collar? (L110-111).

How many samples of for root biomass in each plot and you should supply the diameter of drill sampler (L118-119).

L143. “e” should be italic.

Authors should provide detailed information on how they calculated the Q10 for each month, using the daily mean RS or others.

The α level should be provided for statistics.

Validity of the findings

Why the RS peaked in different months across the three grazing systems (L172-173)? This needs an explanation.

Authors showed that aboveground biomass increased during the experiment, L190, but with what? Sampling period or grazing systems? It should be rewritten and clearly.

I disagree with the authors’ perspective that grazing increased Q10 directly (L222). Because grazing is a method, it influences soil respiration and Q10 through changing environment, such as soil microclimate, photosynthesis, root exudates, soil organic carbon, microbes……

Therefore, the SEM (Fig. 6) should be revised. I suggest the authors to put grazing as a treatment, which changes AGB, soil T, soil M (and other variables), and further lead to changes in soil respiration and Q10.

Authors used “grazing systems”, “grazing sites”. The terms should be consistent throughout the manuscript.

L236-237 Why the rhizosphere respiration in higher in grazing systems than in grazing exclusion systems. This study did not partition components of RS.

L242-244 Authors concluded that DOC exerted important role in soil carbon mineralization, but this is not caused by the positive effects of grazing on DOC. This sentence should be rewritten.

L255 Soil microbial community is affected by environmental variables. You should discuss the change of F: B regarding soil properties or others under grazing intensity and restoration.

L269-271 What positive factors or processes? You should add explanation, such as substrate diffusion.

L276 Generally, there is comparability when site characteristics (the latitude, climate, vegetation type, topography, et al) are comparable between two study sites.

L286-287 It is ambiguous, please rewrite.

L308 The enzyme activity is correlated with temperature sensitivity of SOM decomposition, maybe you can get something from the below paper:
Li et al. Hydrolase kinetics to detect temperature-related changes in the rates of soil organic matter decomposition, 2017; https://doi.org/10.1016/j.ejsobi.2016.10.004

Additional comments

All comments are shown above.

Reviewer 3 ·

Basic reporting

This is an interesting manuscript, which explored the effects of grazing on soil respiration of typical steppe in Inner Mongolia, China. I recommend to accept this paper for publication after minor revision.

Experimental design

The experimental design of the study is scientific and rigorous, the method is sufficiently detailed, and the statistical analysis is reasonable.

Validity of the findings

A better understanding of the effects of grazing on Rs has practical implications for ecological restoration in typical steppe.

Additional comments

In general, the experimental design of the study is scientific and rigorous, the method is sufficiently detailed, and the statistical analysis is reasonable. What’s more, the study is very interesting since it is a field study that analyze interesting properties related with climatic conditions and several microbiological measured in a laboratory some years after the implantation of several management systems as well as during a vegetation growing cycle, which is quite unusual. However, I still have the following concerns.
1. In lines 76-79, three research hypotheses have been proposed. The author should provide more research progress on the above introductions for the second research hypothesis.
2. In the SEM model of Figure 6, the author based on which research foundation to select these factors into the model?
3. In lines 127-129, according to Wu J et al. (1990), the MBC value should be calculated as the difference in carbon content divided by the conversion factor of 0.45 rather than multiplied by it. The author should check if the description was wrong or the data was calculated incorrectly.
4. Figure 4 showed linear relationships between soil respiration and root biomass from December 2014 to September 2015. Why the author only described the linear correlation of the growing season in lines 194-195.
5. In line 206, “Total PLFAs and each different kind of PLFAs all increased over the 2015 growing season in all grazing systems”, does the increase mean that it changes with what? Time change or increase in grazing intensity? Please state clearly and the author should also pay attention to other places in the text.

---

## Round 0.2 · Minor Revisions

As one of the Section Editors for this part of the journal I have taken over this decision as your original Editor is out of contact and unable to make a decision.

There are still some revisions that are necessary (see remarks by reviewer 2). Please re-write your manuscript, taking this reviewers' comments into careful consideration.

Reviewer 1 ·

Basic reporting

ok

Experimental design

ok

Validity of the findings

ok

Additional comments

better revision with good responses

Reviewer 2 ·

Basic reporting

Literature: the authors cited too many literature either from the local region or from Chinese colleagues. They should cite more classic and relevant work on soil respiration and grazing.

Language: it need further improvement.

Implications of the study for better mananing the temperate grassland need better clarification.

Experimental design

Grazing: there are two grazing treatments in the study (yearlong vs. rest-rotation) in addition to the no grazing control. When the authors mention "grazing" in the analysis or writing, they need to be specific - what type of grazing you mean?

For example, hypothesis sections, do both grazing treatments have the same effects?

Validity of the findings

The novelty of the study is still not explained or highlighted enough. Also, limitations of the study (e.g. short experimental duration, no replicate within each plot, problems with Rs partitioning) may be specified in a paragraph, with suggestions for further improvement.

Additional comments

L380: Zhu and Cheng 2011 – it should be the Global Change Biology paper.

Figure 1: x-axis should be a continuous variable (day), not the categorical variable (month). Note that the measurement interval is not fixed. What do error bars mean?

Figure 2: title on y-axis is missing.

Figure 4: indicate the measurement month for these dots with three different symbols.

Figure 6: it was unclear what “grazing” means here. The study has three treatments, no grazing, grazing 1 (yearlong grazing), grazing 2 (rest-rotation grazing). The SEM should be re-formulated. May construct a SEM for yearlong and rest-rotation grazing (compared to no grazing) separately?

---

## Round 0.3 · Minor Revisions

Hi, I made quite a few minor changes to the manuscript to improve the language. I would like you to read through it quickly and make sure that all of my changes are fine (I am hoping none of them change what you were trying to say!). If they are ok, you can accept them and we can accept your paper formally. Thank you and congratulations!

---

## Round 0.4 · accepted · Accept

Thank you for making those changes. I think your paper now reads very well. Congratulations!